

# CDIAC-FF: Global and National CO₂ Emissions from Fossil Fuel Combustion and Cement Manufacture: 1751-2017

5    Dennis Gilfillan[1,2], Gregg Marland[1,2]

[1] Research Institute for Environment, Energy, and Economics, Appalachian State University, Boone, North Carolina

[2] Department of Geological and Environmental Sciences, Appalachian State University, Boone, North Carolina

10    *Correspondence to*: Dennis Gilfillan (gilfillanda@appstate.edu)



**Abstract.** Global and national scale inventories of carbon dioxide ($CO_2$) emissions are important tools as countries grapple with the need to reduce emissions to minimize the magnitude of changes in the global climate system. The longest time series dataset on global and national $CO_2$ emissions, with consistency over all countries and all years since 1751, has long been the dataset generated by the Carbon Dioxide Information and Analysis Center (CDIAC), formerly housed at Oak Ridge National Laboratory. The CDIAC dataset estimates emissions from fossil-fuel combustion and cement manufacture, by fuel type, using the United Nations energy statistics and global cement production data from the United States Geological Survey. Recently, the maintenance of the CDIAC dataset has been transferred to Appalachian State University, and the dataset is now identified as CDIAC-FF. This paper describes the annual update of the time series of emissions with estimates through 2017; there is typically a 2 to 3 year time lag in the processing of the two primary datasets used for the estimation of $CO_2$ emissions. We provide details on two changes to the approach to calculating $CO_2$ emissions that have been implemented in the transition from CDIAC to CDAIC-FF: refinement in the treatment of changes in stocks at the global level, and changes in the procedure to calculate $CO_2$ emissions from cement manufacture. We compare CDIAC-FF's estimates of $CO_2$ emissions with other global and national datasets, and illustrate the trends in emissions (1990-2015) using a decomposition analysis of the Kaya Identity. The decompositions for the top 10 emitting countries show that, although similarities exist, countries have unique factors driving their patterns of emissions, suggesting the need for diverse strategies to mitigate carbon emissions to meditate anthropogenic climate change. The data for this particular version of CDIAC-FF is available at https://doi.org/10.5281/zenodo.4281271 (Gilfillan et al. 2020).

**How to cite:** Gilfillan, D., Marland, G., Boden, T. and Andres, R. J.: Global, Regional, and National Fossil-Fuel CO2 Emissions: 1751 - 2017, https://doi.org/10.5281/zenodo.4281271, 2020



## 1 Introduction

Monitoring emissions of carbon dioxide ($CO_2$) to the atmosphere from fossil fuel combustion and other industrial processes is necessary due to the role of $CO_2$ emissions in driving anthropogenic climate change, and because of the importance and prospects for reducing emissions. Emissions of $CO_2$ impact climate systems, ecosystems, and human systems. Fossil fuel $CO_2$ (FFCO$_2$) emissions inventories are important tools as nations, corporations, and individuals grapple with deciding appropriate reduction targets, and as verification that these reductions are occurring. The global carbon cycle is directly influenced by FFCO$_2$ emissions, and periodic updates through emissions inventories provide information concerning the magnitude and extent of these impacts (Friedlingstein et al., 2019). Information from FFCO$_2$ emission inventories reveals whether emissions are increasing or decreasing, which parties are driving these trends, and what fuel types and economic factors are contributing to emissions.

Current FFCO$_2$ inventories are compiled using data from the production, consumption, and trade of fossil fuels. Data concerning production and consumption are assembled by multiple national and international agencies: the United Nations (UN), the International Energy Agency (IEA), the United States Energy Information Administration (EIA), and BP company being prominent (Andres et al., 2012; Hutchins, Colby, Marland, & Marland, 2017). Depending on the emissions inventory focus, this fossil fuel data can be used to estimate $CO_2$ emissions by fuel type (solids, liquids, and gases) and/or for economic sectors (energy, transportation, manufacture, etc.). Some inventories may also include emissions from additional industrial processes that emit $CO_2$, such as cement manufacture, or emissions from the flaring of natural gas.

Emissions of $CO_2$ from fossil fuel consumption are seldom measured directly, except in recent years at some power plants and other very large point sources, (e.g. (United States Environmental Protection Agency, 2018). FFCO$_2$ emissions are generally estimated from the amount of carbon-based fuels that are consumed. Cement manufacture is often included in $CO_2$ inventories because it is the largest industrial process leading to $CO_2$ emissions that does not involve combustion or the oxidation of non-fuel hydrocarbon products (Gibbs et al., 2000). Cement manufacture emits $CO_2$ into the atmosphere through the process of converting calcium carbonate to lime, an essential ingredient of cement. The FFCO$_2$ emissions from fossil fuels used to support cement manufacture are already included in $CO_2$ emissions inventories (Andres et al., 2012; Andrew, 2019; Le Quéré et al., 2018). Although other industrial processes discharge $CO_2$ into the atmosphere, e.g. iron and steel production, they are often not currently included in emissions inventories because of incomplete data and the recognition that their quantities are generally less than the uncertainty associated with FFCO$_2$ emissions (Andres et al., 2012). Natural gas flaring occurs as a byproduct of petroleum and natural gas extraction and processing, such as in oil fields that are not well connected to natural gas markets, and the related $CO_2$ emissions are often included in global and national inventories.

Although the ultimate goal of inventories is record keeping of FFCO$_2$ emissions, the foci, boundary conditions, assumptions, and initial data sources make each of the currently existing inventories unique. Inventories can also differ on how to deal with fuel used in international trade (bunker fuels), which industrial processes are included, and sometimes even which countries are included. However, consistency within a dataset is important, and changes to any of these aspects with time or place needs to be noted. It is also important to realize that while each



of the current inventories presents estimates of emissions of $CO_2$ for global, regional, and/or national totals, the independent verification of emissions is not presently possible. Estimates are based on survey data, derived average values, and large quantities of compiled data. Space-based monitoring may eventually provide independent, third-party verification.

The longest, most consistent time series dataset on $CO_2$ emissions has long been the time series of global and national emissions generated by the Carbon Dioxide Information and Analysis Center (CDIAC) at Oak Ridge National Laboratory (ORNL) (Andres et al., 2012; Marland & Rotty, 1984). The CDIAC emissions dataset extends from the beginning of the industrial era (1751) to essentially the present, and estimates emissions from fossil-fuel combustion and cement manufacture for all countries (Andres et al., 2012; Friedlingstein et al., 2019; Le Quéré et al., 2018). The CDIAC annual inventories began in 1984 when global interest in $CO_2$ emissions was limited to the scientific community, although focused estimates of global emissions had been produced earlier (Keeling, 1973). The CDIAC emissions estimates are based largely on energy statistics from the UN Statistics Division (United Nations, 2019). The time requirement for the international data collection and processing are such that the UN releases this annual database on a two to three year time lag, which is subsequently reflected in the timeline of the CDIAC $FFCO_2$ emission estimates.

The CDIAC $FFCO_2$ inventory has a cosmopolitan user base; it is currently integral in the Global Carbon Project's annual carbon budget (Canadell et al., 2007; Friedlingstein et al., 2019; Le Quéré et al., 2018), has provided data for the Intergovernmental Panel on Climate Change (IPCC) periodic reports, informs deliberations within the UN, and is utilized by the public and the media as a comprehensive resource for trends in $CO_2$ emissions. However, the United States Department of Energy (USDOE) ceased support for this service at ORNL in 2017. The last release supported by the USDOE included emissions estimates for the year 2014 (Boden, Marland, & Andres, 2017).The CDIAC $CO_2$ emissions time series has been restored in 2019 with independent support from Appalachian State University. The most recent update (through 2017) is the focus of this paper. The historic emissions data from CDIAC at ORNL are stored at the USDOE's Environmental Systems Science Data Infrastructure for a Virtual Ecosystem (ESS-DIVE) data repository at the Lawrence Berkeley National Laboratory. CDIAC at ORNL supported a plethora of additional carbon related research, but this revival is aimed solely at the important dataset of $CO_2$ emissions, so the Appalachian State University initiative is identified hereafter as CDIAC-FF.

Decomposition analysis is an important tool that can be used to characterize temporal drivers of $CO_2$ emissions, addressing issues such as why certain developed countries are declining in emissions (Le Quéré et al., 2019), assessing the socioeconomic aspects of emissions (Pui and Othman, 2019), or identifying drivers of emissions in specific countries using a variety of decomposition techniques (Brizga et al., 2014; O'Mahony, 2013). The most commonly used approach for this kind of analysis with regard to $FFCO_2$ has involved the Kaya Identity, which relates $FFCO_2$ to four primary factors: population, per capita gross domestic product (GDP) (wealth), energy used per unit of GDP (energy intensity of the economy), and $CO_2$ emitted per unit of energy used (carbon intensity of the energy system) (Kaya, 1989). The IPCC has used the Kaya identity to support analysis of emissions scenarios (Pachauri et al., 2014), although much of their focus on reducing emissions has been on the two elements of energy consumption and carbon intensity. While the Kaya Identity has its limitations, it has regularly been employed due



to the availability of quality data and its clear messages and general simplicity (O'Mahony, 2013; Pui and Othman, 2019).

In this paper we first review the methodology to produce the CDIAC-FF emissions estimates (section 2.2) and identify changes that have been implemented in the transition from ORNL to Appalachian State University (Boden et al., 2017; Marland & Rotty, 1984). Two significant changes are noted: the method of including data on stock changes for calculating global totals of $CO_2$ emissions (section 2.2.1) and the approach for calculating $CO_2$ emissions from the production of cement (section 2.2.4). We also discuss trends in the 2017 time series of $CO_2$

(section 3.1) and compare our estimates to other available global inventories (section 3.2). Further, we decompose the Kaya Identity for the top 10 emitting countries to illustrate the drivers of emissions trends from 1990 to 2015 (the end date dictated by the availability of necessary supporting data) and the challenge that different countries face in making significant reductions in emissions (section 3.3).

**2 Materials and Methods**

**2.1 Other global data sets of $CO_2$ emissions from fossil fuel combustion**

    There are currently available four other prominent, annual, global FFCO2 emissions inventories that are "primary" emissions databases. This means that, like CDIAC-FF, the estimates are derived directly from energy data sources. There are also secondary inventories that synthesize

their estimates from multiple primary sources (Andrew, 2020). These primary datasets are available from the IEA, EIA, Emissions Database for Global Atmospheric Research (EDGAR), and the BP Statistical Review of World Energy. Andres et al. (2012) provide a brief discussion of their general characteristics and recently Andrew (2020) has provided a more detailed analysis of the similarities and differences of each of these primary and secondary datasets.

The IEA estimates emissions for both a reference approach (based on fuel type) and a sectoral approach using their own energy questionnaire for member countries, data sharing with the UN for most other countries, national statistical publications, the best estimates from IEA staff experts, and follows the IPCC guidelines for emissions inventories (Andres et al., 2012; IPCC, 2006; IEA, 2019). The IEA data are for $CO_2$ emissions from the energy sector and do not include

emissions from fossil fuel products that are used for non-energy applications such as lubricants and solvents, do not include emissions from gas flaring or cement manufacture, but do include emissions from bunker fuels in their estimates of global total emissions. Recently the IEA has published estimates of 2019 global emissions within 2 months of the year's end, based on partial-year data plus some national and market data releases (IEA, 2020).

The EIA collects their own energy statistics from annual, national-level reports from countries; and uses an approach similar to the approach of CDIAC-FF (Andres et al., 2012). They use internally generated data on the carbon content of fuels and estimates of the fraction-oxidized coefficients in their calculations (Andres et al., 2012; EIA, 2019). EIA inventories do include bunker fuels in national totals, along with emissions from gas flaring and adjustment for non-fuel

uses, but do not include cement manufacture.

    EDGAR is produced as a joint effort of the Joint Research Centre of the European Commission and the PBL Netherlands Environmental Assessment Agency. EDGAR uses the energy balance statistics of IEA in a sectoral approach using the IPCC guidelines for emissions estimates, and represents the emissions from bunker fuels, gas flaring, cement manufacture, and non-fuel uses



using tier I IPCC methods (Andres et al., 2012; Crippa et al., 2019; IPCC, 2006). Note that all of the studies that estimate emissions from cement production rely on cement data from the United States Geological Survey (van Oss, 2019).

The BP Statistical Review of World Energy is the most current $FFCO_2$ inventory, with estimates of emissions reported up to the most recent complete calendar year (BP, 2020). Their estimates
for the two most recent years are often used by other inventories to extrapolate emissions values for the two most recent calendar years (Myhre et al., 2009). This allows the Global Carbon Project, EDGAR, and other $FFCO_2$ spatially-explicit inventories to report more-current estimates of global $FFCO_2$ for researchers and the public (Crippa et al., 2019; Friedlingstein et al., 2019; Oda & Maksyutov, 2011;Oda, Maksyutov, & Andres, 2018). The BP dataset uses IPCC
emissions factors but only considers fuels for combustion, with no distinction for bunker fuels and no other industrial processes (BP, 2020).

## 2.2 CDIAC-FF fossil fuel $CO_2$ emissions estimates

### 2.2.1 Global fossil fuel $CO_2$ emissions

CDIAC-FF uses the UN energy statistics, collected in an annual questionnaire to all countries, to
estimate $CO_2$ emissions (UN, 2019). The information contained in the UN dataset includes production, imports, exports, and changes of stock for all fuels used for energy and non-energy uses. The UN also includes data on fuels that are used in international commerce, known as bunker fuels, and for fuels not categorized as fossil fuels, e.g. wood and other biofuels. Biofuels are not included in estimating $CO_2$ emissions from fossil fuel combustion. The UN period of
record dates from 1950 to essentially the present, with a two to three-year time lag between the initiation of collection and final publication of each year's data. This is a dynamic dataset in which changes, additions, and deletions occur with each annual update of the energy statistics, based on reporting from each individual country. CDIAC-FF is a reference approach to $CO_2$ emissions, meaning that we are focused on emissions from different types of fuel rather than
from different economic sectors. We estimate emissions for three fuel types (solids, liquids, gases) as well as for gas that is flared and for cement manufacture. $CO_2$ estimates based on fuel type facilitate tracking mass flows among parties and makes possible ancillary estimates such as flows for C isotopes (Andres et al., 2000)

Some key differences exist between the approach for estimating the global total of fossil fuel
emissions and for estimating national totals. Fuel production data have traditionally been used by CDIAC for global totals, whereas consumption data have been the standard for estimating national totals. The reason for this is the reduced uncertainty in production data at the global level; fewer data points are needed to calculate production totals rather than consumption totals. Calculations for $CO_2$ emissions are conceptually simple and are the product of three terms: the
amount of fuel i produced ($P_i$), the carbon content of the fuel ($C_i$), and the fraction of the fuel that is oxidized each year ($FO_i$) (Eq. 1). Units for $P_i$ and values used for $FO_i$ and the $C_i$ for each fuel type are summarized in Table 1.

$$CO_2 (as\ C) = P_i FO_i C_i \qquad\qquad\qquad (1)$$

A consequence of using fuel production data to estimate global total $CO_2$ emissions is that all
non-energy uses of fossil fuels are included in the global totals, as are bunker fuels. At the national level, however, we deal with issues of trade, the portion of fuels used outside of national borders, and fuels that are not oxidized. National totals need to estimate the amount of fuel



products that go into long-term products and specifically exclude fuels used in international
commerce. A correction factor (part of $FO_i$ in Eq. 1) is included in the global total calculation to
account for the effective fraction of fuel production that is not oxidized in the year of production
because of sequestration in long-lived, non-fuel products, i.e. we estimate that, on a global
average, 6.7% of the carbon in liquid fuels produced in a given year is sequestered in long-lived
products (Marland & Rotty, 1984). This implies that the balance between the production of
long-lived products in any year and the oxidation of long-lived products produced in earlier years
is such that the total amount of fuels sequestered in long-lived products increases by 6.7% of
annual production.

**Table 1.** Units in primary data source and calculation assumptions for fossil fuel combustion
$CO_2$ emissions estimates. TJ=Terajoules ($10^{12}$ J), tC=metric tons of carbon, tce=tons of coal
equivalent, MtC=Megatons of Carbon ($10^6$ tC)

| Emissions source | Transaction units from UN | Fraction Oxidized $FO_i$ | Carbon Content $C_i$ |
|---|---|---|---|
| Solid fuels | Metric tons[a] | 0.982 | $0.7374 \frac{tC}{tce}$ (hard coal) $0.768 \frac{tC}{tce}$ (brown coal) |
| Liquid fuels | Metric tons | 0.918[b] 0.985[c] | $0.855 \frac{tC}{t}$ |
| Gas fuels | TJ | 0.98 | $13.7 \frac{MtC}{TJ}$ |
| Gas flaring | TJ | 1.00 | $13.45 \frac{MtC}{TJ}$ |

[a] Metric tons are converted to energy units in tons coal equivalent where 1 tce = 2.937x$10^{10}$
joules.
[b] The fraction of oxidized liquids fuels used from global totals
[c] The fraction of oxidized liquid fuels when non-fuel uses are subtracted out for national totals


In the 2016 update to the time series we implemented a change in our computation for the
estimation of the global total of $FFCO_2$ emissions. All CDIAC data sets prior to the CDIAC-FF
data set for 2016 have used only production data, with a global-average value for $FO_i$, for the
estimation of global total emissions for solids, liquids, and gases, as well as for emissions from
gas flaring. However, the 2016 UN energy statistics revealed a substantial drawdown of fuel
stocks already produced and on hand, especially for the solid fuels, and this inspired a refinement
of the CDIAC-FF calculation. Historically, reporting of changes in stocks to the UN Statistics
Division has been such that the data could be used for some countries but were incomplete for
use on total global stocks. The assumption, in essence, was that at the global level there was no
net change in stocks each year.

The reporting of stock change transactions in the primary UN energy data has been increasing
with time and is now judged complete enough to use in the global $FFCO_2$ emissions estimates -
while maintaining consistency with historic estimates. The data show two years in which the
abundance of reported data on stock change transactions increased notably in richness – 1970
and 1992 (Fig. 1a). By 1992 the data on stock changes approaches the completeness seen in
recent year accounts - and this also is the point at which the dissolution of the Soviet Union had
occurred, the unification of Germany was complete, and the array of countries in the dataset was
stabilizing. Thus, inclusion of stock changes is now part of the estimation of global $CO_2$



emissions going back to 1992. Figure 1b shows the quantitative impact of including changes in
stocks in the estimation of annual, global-total $CO_2$ emissions. While 2016 was a noteworthy
year in which inclusion of changes in stocks resulted in a significant increase in the global
estimate of fossil fuels consumed, there are other years where this is also a noteworthy effect. A
net increase in global stocks on hand leads to an overestimate of emissions if stock changes are
not included in the computation, and an underestimate of emissions when global stocks are
decreasing. The average of total global emissions with the change in stocks included (from 1970
to 2017), as compared with global total emissions from production data alone, is 0.26% lower.
This shows that the quantity of stocks in hand has not been changing substantially from year to
year, but is, on average, increasing slowly over time. It is therefore important that the global
emissions time series now includes changes in stocks, and this is reflected in CDIAC-FF
emissions estimates.

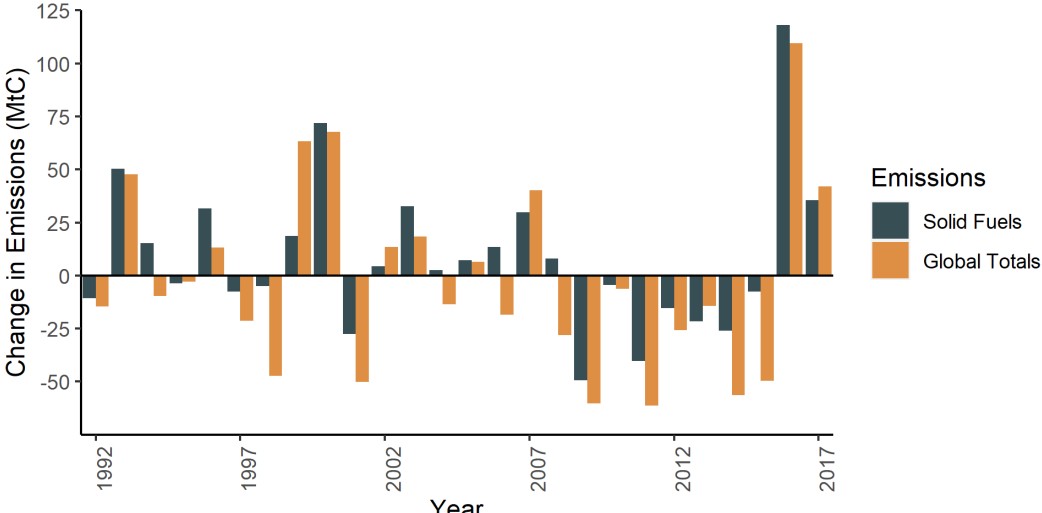

**Figure 1. The change in estimated global total $CO_2$ emissions by including changes in
stocks as opposed to just using production data, in million tons of carbon (MtC). In 2016,**
**the change in global total emissions (orange) corresponds to a 1.10 % underestimation of
emissions if drawdown of stocks is not included in the calculation of global total emissions.
This is mostly attributable to changes in stocks of solid fuels (purple), where including the
change in stocks results led to an increase of 3.15% in emissions from solid fuels. Negative
values indicate that there was an increase in stocks on hand and that $CO_2$ emissions would**
**be overestimated if stock changes were not included. We concluded that data on changes in
stocks were sufficiently comprehensive to be included in calculations of $CO_2$ emissions after
1992.**

### 2.2.2 National fossil fuel $CO_2$ emissions

Fuel consumption data are more informative than fuel production data for scales smaller than
global totals because local specificity is needed to properly allocate emissions. At the national





level fuel consumption (Eq. 2) is estimated using apparent consumption ($AC_i$) and is substituted for $P_i$ in Eq. 1. Apparent consumption is defined as:

$$AC_i = P_i + I_i - E_i - B_i - NE_i - SC_i \tag{2}$$

Where $P_i$ represents production for a given fuel type $i$, $I_i$ represents imports, $E_i$ represents exports, $B_i$ represents bunker fuel loadings, $NE_i$ represents non-energy uses that are unoxidized (assumed to be zero for solids and gases), and $SC_i$ represents stock changes. $CO_2$ emissions from bunker fuels are thus included in estimates of global total emissions but not included in the country totals except to designate the country where fuel loading took place. Emissions of $CO_2$

will occur along international shipping lanes, not in the country where fuel loading took place. Non-energy (non-fuel) uses involve fuel commodities that are used for applications that are not directly consumed for energy uses; examples would be petroleum liquids used to make plastics, lubricants, and asphalt or fertilizer production using natural gas. When the sum of emissions from all country totals does not equal the global total, there are three primary reasons; emissions

from bunker fuels are included in the global, but not in national, totals; emissions from fuels produced for non-energy uses are estimated in the global total, but at the national level non-energy uses are explicitly subtracted out for liquids before estimation of $CO_2$; and the sum of imports for all countries does not equal the sum of exports globally because of statistical errors and incomplete reporting.

**2.2.3 Per capita emissions**

The CDIAC-FF dataset includes estimates of $CO_2$ emissions per capita from 1950 onward. The UN World Population Prospects data are used for global and national level calculations (United Nations Department of Economic and Social Affairs - Population Division, 2020). The projections are produced annually by the UN population division, and we use the standard

projections of population.

**2.2.4 Global and national emissions from cement manufacture**

The manufacture of cement involves calcining carbonate rock, e.g. limestone, to produce CaO-rich clinker, a primary ingredient in cement production. The production of clinker through calcination is one of the largest non-fossil fuel combustion sources of $CO_2$ emissions. The

clinker is then fine ground with gypsum and sometimes other additives to produce finished cement. Calculations based on cement production were, and still are, facilitated by a global database of cement production by country maintained initially by the U.S. Bureau of Mines and subsequently by the USGS (van Oss, 2019).

The biggest change in CDIAC-FF is in the estimates of $CO_2$ emissions from cement manufacture.

The CDIAC emission factor for $CO_2$ from cement manufacture has remained constant and time invariant since 1987, with the assumption that all hydraulic cements had a high proportion of clinker (90-95%). Since that time, however, the quantity of additives in blended cements has increased broadly, that is the fraction of clinker in finished cements has decreased as additives such as coal fly ash and blast furnace slag have increased (Ke et al., 2013; Kim and Worrell,

2002). This made it clear that the original CDIAC methodology was overestimating $CO_2$ from cement manufacture, especially from China, which now produces over half of the world's cement (van Oss, 2019), and required a revaluation of the assumptions for our calculation.



Since the clinker content of cement has been declining since before 1990, and varies with time and place, it follows that the best practice for calculating $CO_2$ emissions from cement
manufacture should be based on the amount of clinker in finished cements (IPCC, 2006). The availability of good data on clinker production or the clinker content of cements really begins in 1990, so we have updated $CO_2$ emissions estimates back to 1990 for the recent edition of the CDIAC-FF time series of emissions. To provide estimates of $CO_2$ emissions from cement production that are transparent and consistent over time and space we rely, when possible, on
clinker-production data that are publicly available and likely to be updated regularly (Case 1). Where data on clinker production are not available we rely on data for cement production and best estimates of the clinker to cement ratio (Case 2). Emissions of $CO_2$ from cement production, $E_{cement}$, are calculated as follows:

$$\textbf{Case 1: } E_{cement} = \frac{M_{CO_2}}{M_{CaO}} r^{CaO}_{clinker} M_{clinker} \tag{3}$$

$$\textbf{Case 2: } E_{cement} = \frac{M_{CO_2}}{M_{CaO}} r^{CaO}_{clinker} r^{clinker}_{cement} M_{cement} \tag{4}$$

Where $\frac{M_{CO_2}}{M_{CaO}}$ is the molecular weight ratio of $CO_2$ to CaO, $r^{CaO}_{clinker}$ is the ratio of CaO in clinker (64.6%), $r^{clinker}_{cement}$ is the clinker ratio, $M_{clinker}$ is the mass of clinker produced, and $M_{cement}$ is the mass of the cement produced. Since the advent of widespread national reporting of greenhouse
gas emissions to the United Nations Framework Convention on Climate Change (UNFCCC) many countries have been reporting values for clinker production in their National Inventory Reports. Time series of clinker production back to 1990 are now available for 31 countries in these National Inventory Reports, and we use this clinker production data to calculate emissions in case 1. We also adopt the IPCC (2006) addition of 2% for cement kiln dust that is not captured
in the cement product to generate a final emission factor ($\frac{M_{CO_2}}{M_{CaO}} * r^{CaO}_{clinker}$) of 0.52 kg $CO_2$ per kg clinker (0.142 kg C per kg clinker).

While cement manufacture is the third largest source of anthropogenic $CO_2$ emissions (after fossil fuel use and land-use change) the availability of the data required for estimating emissions needs improvement (Andrew, 2019). However, for many countries and regions estimates of
$r^{clinker}_{cement}$ are becoming increasingly available. The average $r^{clinker}_{cement}$ globally declined from 83% in 1990 to 78% in 2006, and continued to drop to 67% in 2013, with a rebound after 2013 (Andrew, 2019). The Cement Sustainability Initiative, Getting the Numbers Right, is a global effort to collect environmental data on the global cement industry. It was begun in 2006 by the World Business Council for Sustainable Development and at the beginning of 2019, the work on
the effort was transferred to the Global Cement and Concrete Association (GCCA) (Global Cement and Concrete Association, 2020).

Large quantities of data, including values for $r^{clinker}_{cement}$, are now reported by the GCCA, which we use for individual countries with no clinker production data in National Inventory Reports. There is also an extensive literature on $CO_2$ emissions from cement manufacture in China. From this

publicly available literature we assembled a consistent time-series of the historic $r_{cement}^{clinker}$ for Chinese cement production since 1990 (Cai et al., 2016; Gao et al., 2017; Ke et al., 2012, 2013; Kim and Worrell, 2002; Liu et al., 2015; Shen et al., 2015; Wei and Cen, 2019). The IPCC 2006 inventory guidelines do not endorse the process of calculating $CO_2$ emissions directly from cement production data, but the dearth of international data on clinker production and trade

dictates that using a $r_{cement}^{clinker}$ to estimate clinker production from cement data is often the best choice commonly available.

### 2.2.5 Decomposition of recent $CO_2$ emissions trends

The Kaya Identity, first described by Professor Yoichi Kaya (Kaya, 1989), is a way for us to evaluate factors that drive past and future trends in emissions. The Kaya Identity states that $CO_2$

emissions (C) can be expressed as the product of four terms:

$$C \equiv P * \frac{GDP}{P} * \frac{E}{GDP} * \frac{C}{E} = C_p * C_W * C_{EI} * C_{CI} \tag{5}$$

where P is population, GDP is gross domestic product, and E is primary energy consumption. Data are available from the World Bank on each of these variables(World Bank, 2019). The four factors provide simple representations of population ($C_p$), wealth ($C_W$), the structure and

efficiency of the economy ($C_{EI}$), and the carbon intensity of the energy system ($C_{CI}$). We decompose emissions using a Logarithmic Mean Divisia Index approach (LMDI) (Ang, 2005; Le Quéré et al., 2019), and report relative changes over time in $CO_2$ emissions due to each of the 4 Kaya factors. For the change in C ($\Delta C$) between two given years, in this case year $t_2$ and the reference year $t_1$ , the identity can be decomposed as follows:


$$\Delta C = \Delta C_p + \Delta C_W + \Delta C_{EI} + \Delta C_{CI} \tag{6}$$

Where:

$$\Delta C_x = \frac{C^{t_2} - C^{t_1}}{\ln(C^{t_2}) - \ln(C^{t_1})} \ln \frac{C_x^{t_2}}{C_x^{t_1}} \tag{7}$$

i.e. $\Delta C_x$ is the change in $CO_2$ emissions over the interval $t_1$ (reference year) to $t_2$ which is

attributable to Kaya factor x (Ang, 2005). We decomposed $CO_2$ emissions attributable to each of the factors annually from 1990 to 2015; data were not available to 2017 for each of the World Bank datasets.

## 3 Results

### 3.1 Recent trends in global and national emissions

The global total for $CO_2$ emissions from fossil fuel combustion and cement manufacture in 2017 was 9.79 GtC (Figure 2). After a period of slowing annual growth between 2010 and 2015, the growth rate began increasing again in 2016, with a growth rate of 0.5% in 2016 and 1.2% in 2017. Although all fuels showed an increase from 2016-2017, a 3.1% increase in natural gas emissions was the primary driver of the growth in overall global FFCO₂ emissions. Emissions

from cement manufacture decreased by 1.5% from 2016 to 2017. Since 1990, global emissions have increased by 61.8%, with emissions from solid fuels increasing by 67.2%, liquid fuels





increasing by 37.6%, natural gas increasing by 90.8%, and cement manufacture increasing by 184%. Emissions from solid fuels contribute the most to the 2017 global total (3.94 GtC, or 40.2%), followed by emissions from liquid fuels (3.43 GtC, or 35%), emissions from gases (1.96 GtC, or 20%), emissions from cement manufacture (384 MtC, or 3.9%), and emissions from the flaring of natural gas (76 MtC or 0.7%).

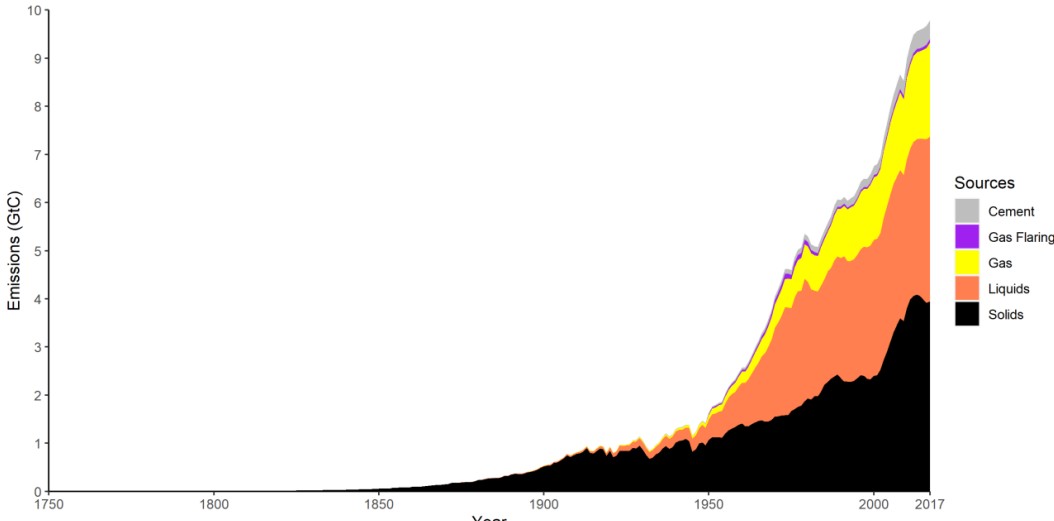

**Figure 2. Total global CO₂ emissions from fossil fuel combustion and cement manufacture from 1950 to 2017, partitioned into fuel type, cement production, and gas flaring. Emissions are in GtC.**

The top 10 emitting countries now collectively emit approximately 65% of the world's total emissions. The top 10 emitters represent countries from North America, Europe, and Asia. These 10 countries' emissions and 2016-2017 growth rates as well as population changes and per capita emissions are summarized in Table 2. China has been the global leader in emissions since 2005 with emissions that have grown by 301% since 1990. The total Chinese CO₂ emissions declined from 2014 – 2016, but saw a 1.7% increase in total CO₂ emissions in 2017. Because of the implications of being such a large emitter of CO₂, accurate accounting is important for Chinese emissions; however, there is uncertainty associated with Chinese data due in part to uncertainty in coal quality and to the improving quality of data on cement (Han et al., 2020).

The country with the largest reported growth in emissions from 2016 to 2017 in the top 10 emitters was Iran, increasing by 21 MtC. This is reportedly driven by a 74 % increase in emissions from the flaring of natural gas (8.9 MtC), followed by a 12.1% increase in emissions from liquid fuel combustion (6.6 MtC) and a 4.9 % increase in the emission from natural gas combustion (5.1 MtC). India's emissions now (2017) are double its 2005 value as it continues to transition as an emergent economy, and the total CO₂ emissions increased by 5.0 % from 2016. Russian emissions are the 4[th] largest in the world, and grew at a rate similar to that of India in 2017. Two countries among the top 10 emitters show decreases in CO₂ emissions from 2016 to 2017 - the United States and Germany. The United States and Germany's decreases are attributed to decreases in solid fuel consumption.





Zambia (37.7%), Mongolia (35.3%), Saint Helena (33.3%), Mauritania (31.65%), and Brunei (26.3%) demonstrated the largest growth rates from 2016 to 2017. The countries that experienced the largest losses in emissions were North Korea (21.0%), the British Virgin Islands (20.3%), United Arab Emirates (18.1%), Ghana (16.9%), and Swaziland (16.4%). These negative values are mostly due to economic downturns/instability, civil unrest, and potential
statistical anomalies.

**Table 2.** Top ten $CO_2$ emitting countries with total $CO_2$ emissions in 2017; population in 2017; the changes in population and emissions from 2016 to 2017; and the 2017 per capita emissions.

| Rank | Nation | Total $CO_2$ Emissions (MtC) | Population (millions) | Emissions change 2016-2017 (%) | Population Change 2016-2017 (%) | Per Capita $CO_2$ Emissions (tC/person) |
|---|---|---|---|---|---|---|
| 1 | China | 2646 | 1421 | 1.67 | 0.49 | 1.86 |
| 2 | United States of America | 1351 | 325 | -0.70 | 0.64 | 4.11 |
| 3 | India | 671 | 1339 | 5.00 | 1.07 | 0.50 |
| 4 | Russia | 494 | 145 | 4.99 | | 3.39 |
| 5 | Japan | 314 | 128 | 0.32 | -0.23 | 2.46 |
| 6 | Islamic Republic of Iran | 198 | 81 | 11.84 | 1.40 | 2.46 |
| 7 | Germany | 196 | 83 | -0.83 | 0.57 | 2.37 |
| 8 | Republic of Korea | 169 | 51 | 0.49 | 0.22 | 3.31 |
| 9 | Saudi Arabia | 156 | 33 | 2.74 | 2.02 | 4.72 |
| 10 | Canada | 156 | 37 | 4.32 | 0.96 | 4.25 |

**3.2 Comparing the different global fossil fuel $CO_2$ emissions inventories**

As noted above, there are currently five primary sources for global estimates of $CO_2$ emissions: CDIAC-FF, IEA, EIA, EDGAR, and BP. These emissions inventories have been prepared by different parties with different objectives, different emphases, different boundary conditions, and different results. Some, for example, include emissions from cement manufacture while some do
not; but we compare the gross reported total of $CO_2$ emissions as included in the respective reports. Comparisons are not simple but we summarize briefly the alternate data sources and the differences that they convey (section 2.1). Figure 2 compares the final estimates of global total emissions for four years (1990, 2000, 2016, 2017), and a sampling of data for six diverse countries that includes the three largest emitting countries.

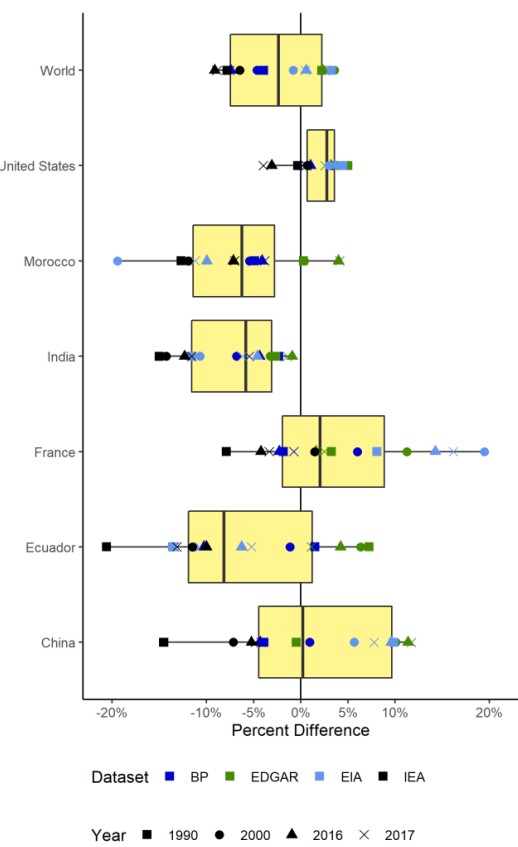


**Figure 3. Comparison of four other global emissions datasets with CDIAC-FF for 1990, 2000, 2016, and 2017. The 0% centerline represents exact agreement with the CDIAC-FF value. Six countries and the global totals were selected to illustrate the variability between datasets and countries. Shapes represent each of the years, and colors represent each of the datasets. Box plots are used to show the general distribution of the percent difference, with the dark line in the box representing the median percent difference, the box representing the range of the 25 and 75 percentiles, and the whiskers representing the overall range of the data. This demonstrates that with few exceptions the estimations are all within 10% of CDIAC-FF estimates for the selected countries and years.**


Although systematic comparison of the alternate datasets has been undertaken (Andrew, 2020; Ciais et al., 2010; Hutchins et al., 2017; Macknick, 2009; Marland et al., 2007; Marland, Brenkert, & Olivier, 1999), the boundary conditions and assumptions used in the calculations make this comparison difficult. Andres et al. 2012 attempted to put them on common ground, and found that the global $CO_2$ emissions agreed to within 3% of the mean (Andres et al., 2012), and this estimate is similar to more recent comparative analyses (Andrew, 2020). Our goal here is to demonstrate a general accord that includes the reinvigorated CDIAC-FF.





Absolute percent differences range from .27% to 20.6% depending on the country, and are less than 10% for the global totals for all four years (Figure 3). At the country level, all of the higher estimates of $CO_2$ emissions (>10%), compared to CDIAC-FF, come from the EDGAR and EIA datasets, while the lower estimates of $CO_2$ ($\leq$-10%) come from the IEA, EIA, and the BP datasets. The larger underestimates are generally from the countries of Ecuador, Morocco, and India, while the larger overestimates, compared to CDIAC-FF, consist of China and France. We suggest that the differences are not indicative of accuracy but rather an indication of the different system boundaries and a measure of the uncertainty. Overall, we estimate that global total emissions have increased by 61.8% since 1990, and from 2016 to 2017 grew by 1.2%. The other datasets report growth from 1990 to 2016 as 56.0% to 62.2% and show a similar growth rate from 2016 to 2017 (1.0% to 1.4%).

Since we have recently updated the procedure for the estimation of $CO_2$ from cement manufacture, it is prudent to also compare the new cement estimates with previous estimates from the ORNL CDIAC, for which the last inventory year is 2014, and a comprehensive global $CO_2$ inventory (Andrew, 2019). Table 2 outlines the total $CO_2$ emissions from cement manufacture for the globe and the top five cement producing countries in each of these datasets. For global totals, ORNL CDIAC estimates grow from 16% higher than these new CDIAC-FF estimates in 1990 to 48% higher in 2014, indicating the overestimation of $CO_2$ emissions because of using the time and location invariant emission factor for cement. CDIAC-FF's global total of $CO_2$ emissions from cement manufacture is within 5% of Andrew (2019). China is a particular country to focus on in this comparison due to its role as the leading producer of cement since 1982. ORNL CDIAC's estimates of $CO_2$ from cement manufacture in China are 34 % higher than the CDIAC-FF estimates in 1990, but this grows to 68% higher in 2014. Much like the global comparisons, Andrew (2019) and CDIAC-FF are within 5% of each other.





Table 2. Comparison of estimates of $CO_2$ emissions from cement manufacture for the globe and
the top five cement producing countries. Data are from the most recent CDIAC-FF update, the
last ORNL CDIAC inventory update, and an independent inventory produced by Andrew (2019).

| Country/World | Dataset | 1990 | 2000 | 2010 | 2014 |
|---|---|---|---|---|---|
| Global total (MtC) | CDIAC-FF | 135 | 188 | 323 | 385 |
| | ORNL CDIAC | 157 | 226 | 446 | 568 |
| | Andrew 2019 | 137 | 195 | 341 | 401 |
| China (MtC) | CDIAC-FF | 21.4 | 61.0 | 159 | 202 |
| | ORNL CDIAC | 28.6 | 81.1 | 248 | 339 |
| | Andrew 2019 | 23.0 | 66.6 | 174 | 212 |
| India (MtC) | CDIAC-FF | 6.1 | 12.0 | 24.2 | 25.2 |
| | ORNL CDIAC | 6.6 | 12.9 | 29.9 | 37.4 |
| | Andrew 2019 | 6.1 | 12.5 | 24.9 | 29.5 |
| USA (MtC) | CDIAC-FF | 8.9 | 11.3 | 8.6 | 10.7 |
| | ORNL CDIAC | 9.7 | 12.2 | 9.1 | 11.3 |
| | Andrew 2019 | 9.1 | 11.3 | 8.6 | 10.8 |
| Turkey (MtC) | CDIAC-FF | 2.9 | 4.1 | 7.9 | 9.0 |
| | ORNL CDIAC | 3.3 | 4.9 | 8.5 | 9.7 |
| | Andrew 2019 | 2.8 | 4.1 | 8.0 | 9.1 |
| Vietnam (MtC) | CDIAC-FF | 0.3 | 1.7 | 6.3 | 6.7 |
| | ORNL CDIAC | 0.3 | 1.8 | 7.6 | 8.2 |
| | Andrew 2019 | 0.3 | 1.5 | 5.8 | 6.3 |

### 3.3 Decomposition of recent trends in $CO_2$ emissions

To gain insight into what is driving changes in $CO_2$ emissions at the country level,
decomposition analysis was performed on the top 10 emitting countries for the period 1990-
2015, or 1992–2015 for Russia and 1991–2015 for Germany. The results are presented as
percentage contributions of the four Kaya-based factors (population, wealth, energy intensity of
the economy, and carbon intensity of the energy system), to $CO_2$ emissions changes based on the
reference year estimates (Fig. 4). For sake of discussion, we will describe positive changes
attributable to a specific Kaya factors as drivers of $CO_2$ emissions, while negative change will be
described as offsets of $CO_2$ emissions.



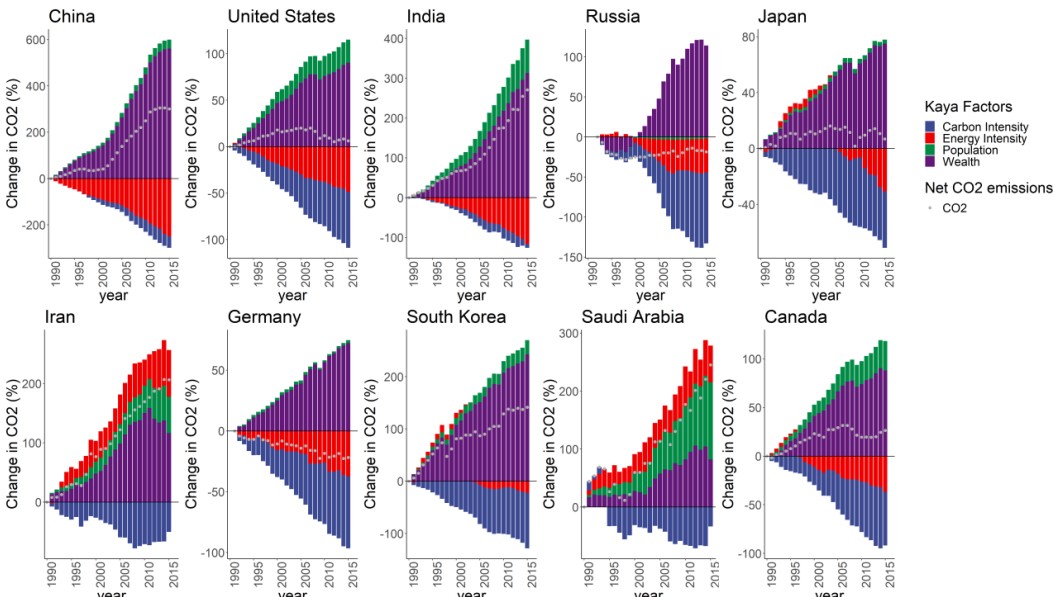

**Figure 4. Log mean Divisia index (LMDI) decomposition of Kaya factors for the top 10 CO₂ emitting countries. The Kaya Factors are outlined in Eq. 3 and the decomposition calculation is outlined in Eq. 4 and Eq. 5. Changes are relative to the reference year 1990 for all countries, except Germany (Reference year 1991) and Russia (Reference year 1992). Positive values indicate drivers of increases in emissions, while negative values indicate offsetting factors. Net CO₂ emissions relative to the reference year are presented by gray dots. The countries are shown in order, from top left to bottom right, of their total CO₂ emissions for the year 2017.**

With the exception of the impacts of the dissolution of the Soviet Union on Russia, increasing wealth (per capita GDP) is a driving force on increasing emissions in each of the top 10 emitting countries. This is especially evident in China, where increasing wealth has contributed to a 561% increase in $CO_2$ emissions from 1990 – 2015. China's growth in wealth is partially offset by decreases in energy intensity (250 % decrease in 2015, relative to 1990). Other countries that see this pattern of increasing wealth substantially driving emissions are India (312% increase 1990 – 2015) and South Korea (243% increase 1990 – 2015). These are emergent, developing economies representing some of the fastest growing economies in the world since 1990. The dominant offsetting factors for these countries are decreasing energy intensity for India (116% decrease) and decreasing carbon intensity for South Korea (106% decrease).

Saudi Arabia and Iran, the top 10 emitting countries from the Middle East, exhibit unique characteristics of the Kaya factors in which energy intensity is a driving force in increasing emissions in addition to population growth and increasing wealth. In Iran, 116% of the growth in emissions from 1990 to 2015 can be attributed to increasing wealth, 79% from increasing energy intensity, and 61 % from population growth. These are modestly offset by decreases in carbon intensity of the energy system (50% decrease). Saudi Arabia is the only nation in the top 10 emitting countries in which population growth is the dominant driving force (132% increase,



relative to 1990 values); decreasing carbon intensity of the energy system only provides modest
offsets (33% decrease) to increasing $CO_2$ emissions.

The remaining top 10 emitters (United States, Russia, Japan, Germany, and Canada) are all
Annex-I countries with obligations to regularly report emissions to the UNFCCC; this potentially
explains the minimal relative growth in $CO_2$ emissions (<50% of 1990 emissions). The countries
are characterized by increasing wealth having the largest magnitude influence on $CO_2$ emissions,
but this is offset by decreases in carbon intensity followed by decreases in energy intensity.
Population growth only contributes minimally to the trends in emissions in each of these
countries, and in some cases (Russia) decreasing population is a small offsetting factor for $CO_2$
emissions.

## 4 Data Availability

The exact version of the CDIAC-FF time series of $CO_2$ emissions from fossil fuel
combustion and cement manufacture that is described in this publication is located here:
https://doi.org/10.5281/zenodo.4281271 (Gilfillan et al., 2020) The historic record of CDIAC
products from ORNL are archived here: https://data.ess-
dive.lbl.gov/view/doi:10.3334/CDIAC/00001_V2017. Future and previous updates from
CDIAC-FF produced at Appalachian State University will be included at https://data.ess-
dive.lbl.gov/view/doi:10.15485/1712447. The most recent inventory year will also be located
within the Appalachian Energy Center's website (https://energy.appstate.edu/research/work-
areas/cdiac-appstate). This includes .csv files for global and national totals as well as a ranking
of each country with regard to total emissions and per-capita emissions for a given year.

## 5 Conclusions

FFCO$_2$ emissions inventories are integral tools to evaluate sources of $CO_2$ emissions, document
trends concerning fuel and/or sectoral-based values, and verify that intended reductions are
indeed occurring. While each of five available global emissions inventories is unique in
approach, focus, boundary conditions, time interval covered, and application; the small
differences in overall emissions estimates demonstrate the accuracy and integrity of the different
products and statistical approaches. Differences do not reflect the degree of accuracy since
independent verification is not currently available at the global and national scales, especially for
$CO_2$ emissions for which there are both natural and anthropogenic sources. CDIAC-FF provides
a long-term time series of FFCO$_2$ emissions that is both comprehensive and consistent over time
and countries. In continuing the CDIAC-FF data set at Appalachian State we provide long-term
continuity while continuing to provide updates and refinements as knowledge and available data
permit. Improving availability of data on stock changes of global fuels and production of clinker
have permitted improved estimates in the 2017 CDIAC-FF dataset.

In addition to evaluating changes in FFCO$_2$ emissions over time, we consider what is driving
recent changes for the top ten emitting countries. To evaluate the possibilities for limiting
emissions in the future it is useful to understand what is driving changes currently. Population
growth, increasing wealth, changes in the energy intensity of the economy, and changes in the
carbon intensity of energy all force emissions in trajectories unique to each country's social
capital and energy resources. Among the top 10 emitting countries, major differences occur in
the balance of forces driving changes in $CO_2$ emissions. For examples, emissions from Germany,
with a net decline in emissions from 1991 onwards, is being driven primarily by changes in





energy intensity while emissions growth in Saudi Arabia is being driven by population growth. The Kaya decomposition approach employed is simple but provides a framework for more extended analysis of the factors driving changes in emissions. While much of the previous 570 analysis on a Kaya framework has focused on energy and carbon intensity, there is a need to characterize the more difficult aspects of carbon mitigation: growth in population and wealth.

The future and equitable confrontation of climate change mitigation will rely on appropriate accounting of $CO_2$ emissions across countries and across time. The top ten emitting countries each have a unique combination of drivers of changing emissions and the need for diverse 575 strategies to mitigate carbon emissions. National and global inventories will provide evidence whether planned emissions reductions are taking place.



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
