# Peer review of "CDIAC-FF: Global and National CO2 Emissions from Fossil Fuel Combustion and Cement Manufacture: 1751-2017"

_Earth System Science Data, 2020_

## Referee Comment (RC1) · Robbie Andrew (Referee) · 4 Jan 2021

Overall: This is an important contribution documenting recent significant changes to the methodology used to construct the CDIAC (now CDIAC-FF) emissions dataset. This dataset is widely used, and thorough documentation is therefore of high importance. The paper is well written, but since many people misunderstand how the dataset is produced and what it includes, I would like to see some ambiguous language tightened, and have made a number of suggestions to help this.

Comments:

[Figure]

Line 18 "combustion": Please be clearer here. Non-combustion emissions are included for solid and gaseous fuels.

Line 41: Ditto.

Lines 44-45: The carbon cycle community has begun to use the more encompassing term "fossil CO2". Consider using this.

Line 49: Update to 2020.

Line 53 "fossil fuels": This excludes carbonate emissions. Please reword.

Line 58 "economic sectors": Be careful with this imprecise term. It means something very different to economists, see Andrew 2020.

Lines 65-66 "does not involve combustion or the oxidation of non-fuel hydrocarbon products": Clumsy and potentially confusing wording. Might be simpler just to say doesn't involve fossil fuels? Further, combustion and oxidation are not mutually exclusive: Combustion is one form of oxidation.

Lines 70-72 "Although other industrial processes discharge CO2 into the atmosphere, e.g. iron and steel production, they are often not currently included in emissions inventories": IEA includes these, EDGAR too. And as I understand it, CDIAC-FF does too (only liquid fuels are adjusted for non-fuel uses, see Andrew 2020). BP does not. I'm not sure "often" is reasonable here.

Line 76 "often included": IEA does not include these.

Line 80 "international trade": 'transport' is better here, since much international trade occurs without transport.

Line 91: "fossil-fuel combustion": It also includes non-combustive oxidation of solid and gaseous fuels.

Line 94 "focused": It's not clear to me what is intended by this word.
Lines 95-96 "CDIAC emissions estimates are based largely on energy statistics from the UN Statistics": Insert words "since 1950".

Line 109 "historic": Replace with "historical".

Line 151 "questionnaire for member countries": Not just members, see Andrew, 2020: "The IEA and Eurostat have developed questionnaires that are sent to at least 61 countries: all members of the Organisation for Economic Co-operation and Development (OECD), European Union (EU), United Nations Economic Commission for Europe (UN-ECE), "and a few others"."

Lines 151-2 "data sharing with the UN for most other countries": Data are obtained from the UN only for small countries. Many more are obtained directly from national statistical publications.

Lines 154-5 "do not include emissions from fossil fuel products that are used for non-energy applications": Be careful here. IEA does include non-energy use in iron and steel manufacture.

Line 156 "do not include emissions from gas flaring": While IEA's main "CO2 Emissions from Fuel Combustion" database does not include emissions from gas flaring, they began to provide separate estimates for flaring emissions in 2020.

Line 169: Also other carbonate decomposition, not just cement.

Line 171 "rely on cement data": Change to "rely at least partly on cement data"

Line 181 "and no other industrial processes": Nor flaring. See Andrew 2020.

Line 187 "international commerce": I think "international transport" is clearer.

Line 202 "reduced uncertainty": I think "lower" is better. Uncertainty is not actively reduced.

Line 205 "fraction of the fuel": More correctly "carbon".

[Figure]

Line 206 "oxidized each year": I'm not sure "each year" is helpful here, and might just confuse people into thinking you have a vintage model, with emissions from a combusted fuel spread over several years.

Line 212 "fuels that are not oxidized": Please be very clear that this only applies to liquid fuels.

Lines 213-4 "fuels used in international commerce": Again, this is highly unclear. Many fuels are traded internationally, and this will give some the impression (or temporarily confuse them) that emissions from traded fuels are not included.

Lines 281-21: This is a little difficult to parse. Perhaps simplify to: This implies that the stock of carbon in long-lived products increases each year by 6.7% of that year's production of liquid fuels.

Line 231 "In the 2016 update to the time series": This will suggest to many readers that this change first appeared in your 2016 release, but you mean the first release to include 2016 estimates. Please reword for clarity.

Line 239 "The assumption": Be clear that this was an assumption made in the CDIAC process, not by the UN.

Line 243 "historic": Change to "historical".

Line 245 "Fig. 1a": I cannot see this figure.

Line 249 "Figure 1b": Should probably be "Figure 1".

Line 282 "assumed to be zero for solids and gases": Please comment on the validity of this assumption.

Line 284: Replace the full-stop with a colon to help the reader understand the connection.

Line 288 "fertilizer production using natural gas": But you have just said that the share

of natural gas not oxidised is assumed to be zero. This is therefore not an example of a non-energy use that is excluded from your emissions estimates.

Lines 299-300 "we use the standard projections of population": It doesn't matter which projection set you use, since they all have the same historical estimates. Please re-word.

Lines 311-12 "the assumption that all hydraulic cements had a high proportion of clinker (90-95%)": According to Andrew 2019, the assumption was a proportion of 98% (p1681). What is the reason for the difference here?

Line 315 "This made it clear": Some references should be made here to studies that made this statement. The earliest were probably those discussing China's cement emissions.

Line 317 "revaluation": Change to "re-evaluation".

Lines 318-319: Provide a reference, or bring forward sentence from 345-347 to here.

Line 328 "are calculated as follows": Provide reference(s).

Lines 336-338: I believe the authors strictly speaking are looking at the Common Reporting Format tables as part of National Greenhouse Gas Inventory (NGHGI) reports submitted by Annex 1 parties to the UNFCCC. There are in fact a number of other countries that officially report clinker production statistics in their National Inventory Reports that are not Annex 1 parties (and do not submit CRFs), and these are used by Andrew, 2019. Please reword.

Line 319 "addition of 2% for cement kiln dust": Please add this to the two equations.

Line 352: "now reported by the GCCA": Some comment should be made on the validity of these data. You don't give a reason for not using GCCA for China. See Andrew 2019, e.g. for China: "The numbers from WBCSD are unreliable because of a very small sample size in China (∼4% of all clinker production), likely to be biased to producers of

higher-quality cement." Similarly, only 47% of India's (second-largest cement producer) production is sampled.

Line 359 "the dearth of international data on clinker production and trade": It is the dearth of data on clinker production that is a reason not to calculate directly from clinker production. But the lack of clinker trade data is a reason not to use cement production statistics. Please reword.

Line 362: Make clear that this is only fossil $CO_2$ (excludes other anthropogenic sources: LUC).

Line 363 "Professor": It isn't necessary or common to give a scholar's position in an academic article, and this usage is inconsistent with all other citations in the present article.

Line 366: Please replace asterisks with multiplication signs.

Line 380 "$CO_2$ emissions": Do you decompose total $CO_2$ emissions from CDIAC, or just those related to energy? It would be sensible to omit cement emissions from this analysis.

Line 381 "for each of": Ambiguous: should this be "any" or "all"?

Line 385 "combustion": Please consider using "oxidation" instead of "combustion".

Line 388 "all fuels": Be clear that you're referring to the three over-arching fuel categories. Perhaps just "all three fuels" is sufficient.

Line 409 "the improving quality of data on cement (Han et al., 2020)": I cannot see any comment in Han et al 2020 about the improving quality of data on cement, so perhaps the citation should be moved earlier in the sentence. The implication of your wording is that data on cement are of better quality in recent years than in earlier years, leading to uncertainty. Is this correct? Are you referring to the availability of clinker production (or ratio) data from 1990?

Line 410: "largest reported growth in emissions": Insert "relative" and report the relative growth here rather than the absolute growth.

Lines 410-1 "reported" & "reportedly": This would be the correct word if you were discussing the energy data, which someone else (UN) reports, but you are discussing the emissions estimates, which you calculate yourselves. I suggest you omit this word.

Line 424-5 "potential statistical anomalies": I suggest adding "particularly for very small countries".

Line 433 "boundary conditions": I'm not sure this is the right term. Boundary conditions is a mathematical term describing what happens when a dynamic system reaches a boundary. "System boundaries" would be preferable here.

Line 462-3 "understimates" & "overestimates": The use of 'underestimates' and 'over-estimates' here implies strongly that you consider CDIAC-FF estimates to be correct and the others are not, but you elsewhere do not give this impression. Please re-word.

Line 463: "India": Andrew 2020b gives some specific reasons why CDIAC estimates are too high for India (Figures 3 and S17: https://doi.org/10.5194/essd-12-2411-2020). For coal, the UN values for carbon content (used by CDIAC, and provided to the UN by India) before 2014 are very likely incorrect. This suggests errors rather than just 'different system boundaries' as you suggest.

Line 477 "within 5% of Andrew (2019)": Some comment might be appropriate. For example, Andrew 2019 sources clinker production for a number of other countries from their reporting to the UNFCCC.

Lines 480-1 "Much like the global comparisons, Andrew (2019) and CDIAC-FF are within 5% of each other.": Some comment on why these might differ is appropriate. For example, Andrew 2019 uses published clinker production data, while the present article uses clinker ratios.

Line 488: This section shows results of some Kaya analysis, but with little interpretation

and no comparison with numerous other studies that do similar analyses. Perhaps this is ok in a data journal, so I will allow the editor to comment here. But in most journals additional interpretation and comparison would be required.

Lines 507-8 "With the exception of the impacts of the dissolution of the Soviet Union on Russia, increasing wealth (per capita GDP) is a driving force on increasing emissions": I'm not sure the exception is required in this sentence. The general rule observed is that as wealth increases, so do emissions. In Russia following the break-up, wealth declined.

Line 517 "top 10 emitting countries from the Middle East": Remove "10".

Lines 527-8 "this potentially explains the minimal relative growth in CO2 emissions": The implication being that they make more effort because they're required to report? I don't think this follows. More likely that these countries accepted targets/reporting requirements because they were past the kuznets peak.

Line 549 "boundary conditions": As above, I suggest instead "system boundaries".

Line 550 "demonstrate": This is too strong. Use rather "suggests" or "indicates". You point to this very issue in the sentence that follows.

Line 545 "comprehensive": It is not entirely comprehensive, since it omits both emissions from non-energy uses of oil products and emissions from decomposition of carbonates other than in cement production. Please reword slightly.

Line 545 "consistent": This should also be worded more carefully. The data source pre 1950 is quite different. Further, while CDIAC-FF uses a consistent approach and data sources for estimates starting 1950, those data sources themselves are produced somewhat inconsistently across countries and time. Consider for example the contrasting case of GCP, whose methods are relatively inconsistent across countries, but if author X were to cite GCP as their data source for further calculations, X could readily say their method and data are 'consistent'. So what is the value of the 'consistent'

description? The value here is that it reduces the burden on producing the dataset, which is good. But I recommend that the authors avoid implying anything more than that.

One last minor comment: While it makes little difference, the authors might consider in a later revision extending the dataset back to 1750. The source used by Andres et al 1994 (i.e. Etemad & Luciani) interpreted a source (Pollard 1980) incorrectly. Pollard provides "quinquennial" estimates (averages over five-year periods) of coal production for Great Britain, but the first period is six years, starting in 1750. This is why GCP's data now start in 1750.

---

## Referee Comment (RC2) · Anonymous Referee #2 · 31 Jan 2021

Review of the paper "CDIAC-FF: Global and National CO2 Emissions from Fossil Fuel Combustion and Cement Manufacture: 1751-2017" by Gilfillan and Marland. The paper presents a useful update of a widely-used global and national CO2 emissions dataset along with some basic analysis of the results. Some aspects of the paper should be revised before publication.

Overall, the methodology is not defined in detail. While the main parameters are defined in Table 1 and the text, the details are basically a black box. This doesn't really meet current open-source standards. Its actually quite difficult to trace the methodology in detail through all the past CDIAC reports and papers. While I doubt the authors

can address this completely, it would be useful if they could discuss this general issue.

Additional specific comments are below.

Line 86 - perhaps mention that space-based validation is complicated by large land-use fluxes of CO2.

Section 2.1

The paper should probably also discuss the CEDS inventory (Hoesly et al. 2018), which also extends from 1750 (using some data in common to CDIAC) and was used in the global CMIP6 climate modeling exercise. CEDS also largely a primary CO2 emissions inventory (energy emissions are calculated using energy consumption and emission factors, although sector emissions are provided) while process emissions are from other sources. The CEDS web site indicates that a new version extending to 2019 may be out soon.

Line 170 - EDGAR also includes emissions from some additional processes (such as lime manufacture and fossil fuel fires) that are not included in the other inventories mentioned.

Lines 205 - 210.

It would be useful to better clarify the issues of fuel oxidation here. There are two pathways for non-oxidized fuels, incomplete combustion (e.g. some carbon in the fuel does not exit the smokestack, e.g., is retained in ash or soot within the combustion devise), and fuel used as feedstocks (e.g., "non-energy uses of fossil fuels" in the author's language) that are subsequently not oxidized.

Line 215 - Clarify (I assume?) that the "correction factor" is applied to the FOi in Table 1 for the global estimate? Clarify if this applied equally to all fuels?

It would be useful somewhere to note that these estimates (like all the others mentioned) represent eventual CO2 emissions into the atmosphere, but actually include

incomplete combustion (BC, OC, CO, NMVOC, CH4) emissions at the point of emission. Implicitly assuming all of these emissions are eventually fully oxidized).

Line 412 - The earlier methodology section does not discuss the data sources for flaring $CO_2$. This should be added. Then at this point, it would be useful to comment on the accuracy of the flaring data from Iran. Is this from remote sensing? I would presume there is significant uncertainty associated with this data? (How does this estimate compare with other sources, given its this source that puts Iran into the top 10?)

Section 3.3 Decomposition The data used for this analysis needs to be better defined. There are many different metrics that can be used for GDP, for example, including different dollar years. Also is MER or PPP-based GDP used, this can make a large difference.

It is not accurate to use the Kaya identity to say that "increasing wealth" has contributed to an increase in $CO_2$ emissions. What has happened in the real world is a complex system where increasing wealth is accompanies by changes in economic structures and applications of new technologies, all of which impact emissions. The Kaya identity is simply one way of decomposing the major driving forces, but they are actually much more entangled than depicted here. Use of the identify is fine, but the discussion needs to be re-written to be more careful about how the results are described. The Kaya identity does not show causation, it is simply a sometimes useful decomposition method.

The authors have not shown that "population growth is the dominant driving force" in Saudi Arabia. What they have shown is that population is the largest factor in the decomposition. It would take an analysis of the structure of emissions growth over this period to show if population growth was actually "the dominant driving force".

(Language in the conclusion section should be similarly edited to be more accurate).

Figure 3 It would be useful perhaps to add a dotted vertical line at +-10%.

[Figure]

The exact versions of each data set used in Figure 3 needs to be specified.

Figure 3, and accompanying discussion, would be useful if there was some attempt to compare like-with-like taking into consideration different system boundaries of the datasets. The authors have not really shown that "We suggest that the differences are not indicative of accuracy but rather an indication of the different system boundaries and a measure of the uncertainty", but this is instead really the conclusion from previous work and should be re-stated as such.

Line 527. I rather doubt that "obligations to regularly report emissions to the UNFCCC" are the reason explaining slow growth in CO2 emissions. (or, at least the authors have now shown this).

Table 2 - It would be useful to extend the table and comparisons out to 2017 given that both the new CDIAC and Andrew exist at this point (and the continuing trends of decreasing clinker ratio). I suggest re-ordering the rows, with ORNL CDIAC first so that CDIAC-FF and Andrew next too each other since they are more similar.

Hoesly, R. M., Smith, S. J., Feng, L., Klimont, Z., Janssens-Maenhout, G., Pitkanen, T., Seibert, J. J., Vu, L., Andres, R. J., Bolt, R. M., Bond, T. C., Dawidowski, L., Kholod, N., Kurokawa, J.-I., Li, M., Liu, L., Lu, Z., Moura, M. C. P., O'Rourke, P. R., and Zhang, Q.: Historical (1750–2014) anthropogenic emissions of reactive gases and aerosols from the Community Emissions Data System (CEDS), Geosci. Model Dev., 11, 369–408, https://doi.org/10.5194/gmd-11-369-2018, 2018

---

## Author Comment (AC1) · 2 Mar 2021

Referee Robbie Andrew

Referee Andrew is recognized as a high level expert and he is a well informed and diligent reviewer. His comments on our paper are incredibly valuable and have caused us to rethink many statements, to correct several mis-statements, and to make other text more precise and clearer. His thoroughness is much appreciated. In the following we summarize how we have responded to his comments. In most cases we have responded exactly, or nearly exactly, as he suggests. In other cases we have rewritten a short section of text to incorporate his suggestions. In a number of cases we feel that his suggestions represent personal preferences and so long as our statement is factually correct we have stayed with our own preferences and we indicate "no action". Occasionally we feel his ideas are adequately dealt with in the cited references and do not need reiteration or expansion in this short paper, and we again note simply "no action". In total we are very aware that the comments of Referee Andrew have led us to a paper that is more accurate and clearer and we are grateful.

Line 18: Since we use the global assumption for non-energy uses for country level totals (0.01 for gases, 0.8% for solids), these industrial processes are not necessarily included in the calculations, see Marland and Rotty 1984, pg. 243 and 249. This was an error in personal communication and we have reiterated it in other places as well.

Line 41: done. Non-energy uses is the term used by the UN, and we are trying to be more consistent with that language.

Line 44-45: done

 Line 49: done

line 53: done

line 58: Although Marland disagrees, Gilfillan agrees that economic sectors is imprecise, and have tried to correct this with "sectors of human activity" as a middle ground.

lines 65-66: done

lines 70-72: no action. See comments from line 18. (1% gases, 0.8% solids) (Marland and Rotty 1984).

line 76: done

line 80: done

line 91: done

line 94: deleted

lines 95-96: done

line 109: done

line 151: done

lines 151-152: done

lines: 154-155: added "The IEA does include some non-energy uses from iron and steel manufacture."

line 156: added "and recently provides separate emissions estimates from flaring emissions not within their main $CO_2$ database."

line 169: done

line 171: done

line 187: done

line 202: done

line 205: done

line 206: done

line 212: no action on this particular phrasing. We apologize for our errors in personal communication, but hopefully we have clarified them in this discussion. The global percentages are used as an assumption of nonenergy uses for solids and gases for country level totals.

lines 213-214: done

lines 281-21: no action

line 231: done

line 239: done

line 243: done

Line 245 and line 249: done

Line 282: sentence has been corrected. "$NE_i$ are explicitly subtracted out for liquids based on the UN energy statistics codes, and we use the global assumptions (section 2.2.1) for the amount of solid and gaseous fuels that are used in for non-energy purposes, 0.8% and 1% respectively."

Line 284: no action

Line 288: the sentence is correct, natural gas is oxidized in the production of fertilizer

Line 299-300: standard rather than probabilistic.

Line 311-312: sentence corrected

Line 315: Sources have been added from Andrew 2018, 2019

Line 317: done

line 318-319: sources added.

line 328: Sources have been added.

lines 336-338: the text is correct as written, Marland in developing his method only used data from NIR/CRFs as far as UNFCCC  no action

line 319: done.

line 352: the Andrew paper is cited twice in this page of text and in the data file, we think that this is excessive detail for this discussion.

Line 359: no action

Line 362: Added "Fossil"

Line 363: this is not a simple citation.  Considering the context we think this is appropriate, no action

Line 366: done

Line 380: We used all emissions in CDIAC, not just combustion of fossil fuels. We will take your suggestion for future Kaya decompositions.

Line 381: no action

line 385: done

line 388: done

line 409: yes, the citation was misplaced.  The sentence has been corrected.

Line 410: done

Line 410-1: done

Line 424-5: done

Line 433: done

Line 462-3: the usage is clear in the context

Line 463: A previously unknown but very timely reference has been inserted. (Thanks Referee Andrew!)

Line 477: appropriate references are cited, no action

Line 480-1: appropriate references are cited, no action

Line 488: we believe that an appropriate level of interpretation is provided in the near-by text, no action

Lines 507-508: no action

Line 517: done

Lines 527-8: sentence deleted

Line 549: done

Line 550: done

Line 545: text has been rephrased

Line 545: text has been rephrased

"one last minor comment": no action expected or provided

---

## Author Comment (AC2) · 2 Mar 2021

Referee #2

Paragraph 2, "Overall the methodology is not defined in detail".

Response:  We have added a sentence to the introduction to emphasize that the details of the methodology are carefully established in the 1984 paper by Marland and Rotty.  The core of the computer code and half of the author team are unchanged after 30 years.

Line 86: added "but are difficult due to fluxes of natural sourced $CO_2$."

Section 2.1. "The paper should probably also discuss…"

Response:  The Hoesly paper is a new (2018) contribution that contributes by providing consistent inventories across multiple species, but it does not provide annual updates and does not meet ours, or Robbie Andrew's, definition of a primary dataset.  It is nonetheless a valuable contribution and we have added a citation in section 2.1.

Line 170.

Response: no action, this is consistent with our mention of carbonate decomposition and our treatment of solid fuels (see Marland and Rotty, p. 248).

Lines 205-210.

Response.  We add another reference to Marland and Rotty 1984 to emphasize that this is dealt with in detail in the earlier paper.  It is also dealt with in the following paragraph.

Text line beginning "it would be useful…"

Response: No action.  We hate to dwell on the 1984 paper by Marland and Rotty but it is clear that Referee #2 is not familiar with this paper and it is clear that the early portion of our current paper does not adequately convey the importance of Marland and Rotty in establishing the methodology that is preserved in this paper (and was very influential in the evolution of the IPCC methodologies).  To this point we have added additional citations to Marland and Rotty and we have tried to better establish its role in describing our methodologies.

Line 215.

Response.  No action, this is also dealt with in detail in the Marland and Rotty 1984 paper, pg. 249. 0.8 % of solids fuels are assumed to be incomplete combustion, and 1 % are assumed to be for non-energy uses FO= (.982). This is included in multiple other places as well. For gases, it's 1% and 1% so FO of 0.98. We treat only liquid fuels differently at the global (6.7% nonenergy uses) and national level (subtract out all nonenergy use codes). Hope that helps in your understanding.

Line 412.

Response:  The UN uses a code 104 to describe natural gas flaring. The global uncertainty for these estimates is described in Andres et al. 2014 (25 % 2 sigma uncertainty for combined uncertainty). We do not use any remote sensing data because we strive to maintain some consistency from the original CDIAC estimates from Marland and Rotty 1984.

Section 3.3

Response:  Phrasing has been added to say "GDP is gross domestic product (Purchasing power parity (PPP), current international dollars)" in section 2.2.5.  We have added a second sentence to emphasize that the factors of the Kaya decomposition are taken as simple representatives of complex concepts. We agree with the statement on Saudi Arabia but believe that our representation of the Kaya identity is clear in context.

Figure 3 and accompanying discussion

Response: "Our goal here is to demonstrate a general accord that includes the reinvigorated CDIAC-FF."

As cited in the text this difficult task has been attempted by Andres et al. and by Andrew and is beyond the intent of this paper. Our purpose here is to demonstrate that we haven't really changed much of our methodology in the transition from CDIAC (Marland and Rotty 1984) to CDIAC-FF, and not to reproduce Andrew 2020's work in his comparison of emissions; just show that they are in general agreement. Dotted lines have been added, and each of the versions used are cited in section 2.1. Text has been added into the figure description about the exact versions of the data products used, and some comparisons with regards to what is included in these data products is described in section 3.2.

Line 527.

Response: sentence has been removed.

Table 2.

Response:  The ORNL-CDIAC data set ends with data for 2014.